# DEXA Body Composition Asymmetry Analysis and Association to Injury Risk and Low Back Pain in University Soccer Players

**DOI:** 10.3390/ijerph21050559

**Published:** 2024-04-28

**Authors:** Nicolas Vaillancourt, Chanelle Montpetit, Victoria Carile, Maryse Fortin

**Affiliations:** 1Department of Health, Kinesiology and Applied Physiology, Concordia University, 7141 Sherbrooke Street W, Montreal, QC H4B 1R6, Canada; vaillancourtn97@gmail.com (N.V.); c_montp@live.concordia.ca (C.M.); 2Concordia Science College, Concordia University, Montreal, QC H4B 1R6, Canada; victoria_carile@hotmail.com; 3School of Health, Concordia University, Montreal, QC H4B 1R6, Canada; 4CRIR—Centre de Réadaptation Constance-Lethbridge du CIUSSS COMTL, Montréal, QC H4B 1T3, Canada

**Keywords:** body composition, dual-energy X-ray absorptiometry, asymmetry analysis, injury risk, low back pain, soccer players

## Abstract

Soccer is a laterally dominant sport owing to the repetitive nature of unilateral kicking. The relationship between functional and body composition asymmetries related to limb dominance in soccer players has yet to be established. When present, asymmetries can increase the risk of injury and low back pain. Our study investigated whether lateral dominance is associated with limb asymmetries in a comprehensive body composition assessment among varsity soccer players. Twenty-seven varsity soccer players (age 20.4 ± 1.7 years old; BMI 22.6 ± 4.6 kg/m^2^) participated in this study. Body composition was assessed through dual-energy X-ray absorptiometry scans. Results showed low lower limb asymmetry indices in both males (3.82%) and females (3.36%) compared to normal ranges. However, upper limb lean mass exhibited high asymmetry, surpassing thresholds in males (7.3%) and females (4.39%). Significant differences were found in total bone mass among males and total lean body mass among females. Male players exhibited higher asymmetry indices in both arm and trunk mass compared to females. Despite these asymmetries, no significant correlations were found between asymmetry indices and occurrences of lower limb injury or low back pain. The study suggests that while evaluating body composition for injury prevention in soccer shows potential, lateral dominance may be influenced by factors extending beyond sport-specific adaptations.

## 1. Introduction

Soccer stands as one of the world’s most popular sports. A survey from FIFA in 2006 estimated that a staggering 265 million individuals worldwide actively play soccer [1]. The popularity and notoriety of professional soccer leagues draw numerous young athletes to specialize in the sport. Soccer, characterized by lateral dominance, presents unique demands to its participants [2]. Lateral dominance, often referred to as laterality, denotes an individual’s inherent tendency to favor one side of the body over the other when performing motor tasks [3]. In soccer, this phenomenon is epitomized as “footedness”, indicating the preferred foot and limb used for ball control and kicking. Lateral limb dominance in soccer players has been attributed to the repetitive nature of unilateral kicking and the heightened frequency of use of their dominant leg during training [4]. Athletes specialized in sports that exhibit lateral dominance may develop functional and body composition asymmetries, a phenomenon well documented in tennis players [5]. 

Despite the unilateral nature of soccer, there is no consensus on the presence of lower limb functional asymmetry in soccer players. The most common method of measuring lower limb functional asymmetry in soccer players is through the measurement of muscle strength. Some studies report greater strength in knee flexion and extension of the dominant limb [6,7], while others suggest greater strength of the non-dominant knee flexors [8], with some reporting no differences [4]. 

When present, lower limb functional asymmetry increases the risk of injury in soccer players. Lower limb injuries (LLI) are highly prevalent in soccer, accounting for 60–90% of all injuries that occur [9,10,11], while low back pain (LBP) emerges as one of the most common overuse injuries in the sport, with a lifetime prevalence of up to 64% [12]. A 2021 systematic review reported lower limb functional asymmetry (i.e., strength and ROM) as a risk factor for injury in soccer players [13]. Significant differences in strength observed during isokinetic testing of the knee and ankle are associated with an increased risk of injury [14,15,16]. Functional asymmetry can also be measured through the Functional Movement Screen (FMS) test. A study evaluating soccer players with the FMS revealed a moderate probability (15%) of injury when at least two FMS asymmetries were detected [17]. However, subsequent studies were unable to reproduce these findings [18]. Authors incorporating at least one FMS asymmetry as part of their injury prediction criteria identified an increase in the probability of injury risk [19,20]. Potential mechanisms suggest that the distinct internal demands of kicking with the dominant mobilizing limb compared to the non-dominant stabilizing limb contribute to an imbalance in force production between limbs [4]. Kicking produces different forces between the limbs, consequently increasing torque around the pelvis [21]. The force distribution may be unevenly dispersed throughout the body and is primarily absorbed by the spine, potentially leading to LBP. Lower limb asymmetries can also cause functional constraints that limit movement strategies [22]. Limited movement strategies can prompt athletes to adopt motor behaviors that increase the risk of LLI [13]. Functional asymmetry is a risk factor that may contribute to the high rates of LLI and LBP in soccer players, thus exploring other components of asymmetry is important to better understand its impact on injury rates.

Body composition symmetry is an important and often neglected component of symmetry in athletes. Studies have reported body composition asymmetries in lower limb lean body mass (LBM) [23] and trunk LBM [24], while others reported no differences in body composition [4]. More evidence is required using gold standard measurement tools for body composition (i.e., DEXA) to better determine whether body composition asymmetries are present in soccer players. Body composition asymmetry may play an important role in predicting injuries among soccer players; however, the specific implications of this asymmetry on injury risk remain largely unexplored. There is some evidence suggesting that asymmetries exist in the trunk musculature of soccer players, such as the internal oblique and multifidus muscles, which are associated with LBP [25,26]. However, no studies have examined the effect of body composition asymmetry on LLI risk in soccer players. Therefore, the global impact of body composition asymmetry on injury risk in soccer players remains poorly understood. Better documentation of body composition asymmetries in soccer players and understanding the impact of such asymmetry on injury rates are vital to develop and implement preventative strategies for athletes. The adaptability of musculoskeletal tissue to mechanical loading, coupled with the characterization of morphological differences in body composition, may provide important modifiable factors for injury screening and facilitate the development of targeted training and rehabilitation programs.

The purpose of our study was to (1) examine body composition asymmetries in university-level soccer players, and (2) determine if body composition asymmetry is associated with LLI and LBP. We hypothesized that (1) significant asymmetry in lower limb and trunk LBM would be observed and (2) lower limb and trunk LBM asymmetry would be associated with an increase in LLI and LBP. 

## 2. Materials and Methods

### 2.1. Study Design and Setting

A cross-sectional study was conducted, and all research activities took place at the School of Health, Concordia University, Montreal, QC, Canada. This study was approved by the Central Ethics Research Committee of the Quebec Minister of Health and Social Services (#CCER-16-17-06). Each participant provided their informed consent by signing a consent form.

### 2.2. Participants

A total of 27 university varsity soccer players (12 females, 15 males) were included in this study, with a mean age of 20.4 years old (range 18–23 years old). Participants were eligible for enrollment if they were a Concordia University soccer varsity players, and excluded if they met any of the following criteria: (1) any previous history of spinal trauma, (2) spinal abnormalities (e.g., scoliosis > 10°, spondylosis, or spondylolisthesis), (3) prior spinal surgery, or (4) pregnancy. 

### 2.3. Body Composition Assessment

Body height measurements were collected using a mechanical wall-mounted system (Perspective Enterprises, Easy Glide Bearing Stadiometer, Portage, USA) and body mass was determined using a digital scale (Rice Lake’s Weighing Systems, Rice Lake, WI, USA). A certified medical imaging technologist conducted a comprehensive full-body DEXA scan (Lunar Prodigy Advance, General Electric HealthCare, Chicago, IL, USA). The DEXA scan is the gold standard of body composition measurements [27]. To prevent any interference with scan accuracy, participants wore loose-fitting clothing and removed any metal accessories. During the scan, participants were in a supine position with their spine centered in the scanner, their arms slightly away from the body, and their legs slightly spread apart to ensure optimal scan precision (e.g., both thumbs and toes pointing upwards). 

Anthropometric and body composition data were used to obtain the following variables: upper limb bone mass, lower limb bone mass, trunk bone mass, total bone mass, upper limb LBM, lower limb LBM, trunk LBM, total LBM, upper limb fat mass, lower limb fat mass, trunk fat mass, total fat mass, total body fat percentage, body mass index (BMI), and asymmetry index (AI). The total bone mass was estimated as the sum of the bone mass from the upper limb, lower limb, and trunk. Total LBM was estimated as the sum of the LBM from the upper limb, lower limb, and trunk. The total fat mass was estimated as the sum of the fat from the upper limb, lower limb, and trunk. The side-to-side AI was calculated using the formula below, a commonly used formula in morphological studies [28,29,30].

(1)
$$Asymmetry\;Index\;\left(AI\right)=\frac{\left|L\;side\;mass-R\;side\;mass\right|}{0.5\times (L\;side\;mass+R\;side\;mass)}\times 100$$


### 2.4. Asymmetry Index (LBM, Bone Mass, Fat Mass) 

Until recently, there were no normative data to distinguish between low and high asymmetry in LBM distribution. In a study published in 2022, Minetto et al. used data obtained from the National Health and Nutrition Examination Survey, collected from 1999 to 2004 by the Center for Disease Control, to establish normative data for the LBM AI [29]. They analyzed a total sample of 10,014 participants and used a subsample of 1915 young and healthy participants aged 18–39 years old (52.9% males and 47.1% females) with a normal BMI (18.5–24.9 kg/m^2^) to produce the reference values presented in Table 1. There was no information available regarding the physical activity levels of the participants. They used the mean to establish a normal AI, with the high range defined as exceeding +2 standard deviations from the mean. A low AI refers to a normal physiological difference whereas a high AI refers to a pathological difference. 

### 2.5. History of Injury

Self-administered questionnaires were used to gather information on players’ injury histories and experiences with LBP. The questionnaire covered essential aspects such as the player’s position, the number of years playing at a competitive level, and the number of years they had been playing with Concordia University. The injury-related questions were specific to the player’s LLI history (e.g., hip, thigh, groin, knee, ankle, and foot) within the past 3 months and 12 months. Participants were asked to specify the injured body part and the duration of the injury.

LBP was defined as pain localized between the twelve thoracic vertebrae (T12) and the gluteal folds, with or without leg pain [31]. Participants were asked if they experienced LBP within the past 4 weeks and 3 months. If they responded “yes”, they provided the duration and recurrence of the LBP, along with specific location details (e.g., centered, right side, or left side). Furthermore, participants were asked to rate the average pain intensity using the Numerical Pain Rating Scale (NPRS), with 0 indicating no pain and 10 being the worst pain imaginable.

### 2.6. Procedure

Body height and mass measurements for each player were taken at the Concordia University School of Health during their preseason assessments. Subsequently, a body composition assessment was conducted to evaluate the player’s fat mass, lean muscle mass, and bone mass. Following the physical assessments, the participants completed a questionnaire to gather information regarding their soccer-related activities, injury history, and history of LBP. 

### 2.7. Statistical Analysis

No a priori sample size calculation was performed for this project. All varsity soccer players (males and females) were invited to participate to maximize the sample size. Descriptive statistics were used to assess the normality of data distribution for each variable. The Kolmogorov–Smirnov test was used to test the normality assumption for the difference between the paired values of each body composition parameter. As the Shapiro–Wilk test showed normal distribution of the data, parametric tests were used, with means and standard deviations used as sample parametrics. To compare the right and left sides for body composition parameters with a normal distribution, paired *t*-tests were conducted. For parameters with non-normal distributions, the Wilcoxon signed-rank test was used. Independent *t*-tests or Mann–Whitney U tests for non-normally distributed data were used to assess asymmetry (expressed as the absolute value difference between the right and left sides) among male and female players. A logistic regression analysis, adjusted for sex, was used to determine the correlation between the AI, LLI, and LBP using an odds ratio. The threshold for statistical significance was set to *p* < 0.05. All analyses were performed using SPSS (version 26.0.0., IBM, New York, NY, USA).

## 3. Results

### 3.1. Demographics

The characteristics (e.g., age, height, and weight), soccer-specific characteristics, LLI history, and LBP history for male and female soccer players are presented in Table 2. The average number of years playing soccer at a competition level was 8.5 years and 1.4 years at the university level. A total of 30% of players reported LBP within the preseason, while 48% reported having had an LLI within the past 12 months. 

### 3.2. Bone Mass, LBM, and Fat Mass Measurements in Male and Female Soccer Players

Body composition characteristics for male and female participants are presented in Table 3. There was a significant difference in bone mass between the right (158.80 ± 25.64) and left (153.25 ± 20.94) upper limb in female players (*p* = 0.02). There were also significant differences in bone mass in male players, with the right upper limb (*p* = 0.02) and right total bone mass (*p* = 0.04) demonstrating greater values. There was a significant difference in total LBM between the right side (21,675.83 ± 2138.11) and left side (21,936.14 ± 2013.13) in females (*p* = 0.02). There was a significant difference in upper limb LBM between the right side (3611.97 ± 673.13) and left side (3421.02 ± 532.90) in males (*p* = 0.04). There was no significant LBM asymmetry for the lower limb or trunk in male or female players. There was a significant difference in fat mass between the right upper limb (1065.76 ± 295.76) and left upper limb (1028.50 ± 277.12) in females (*p* = 0.01)

### 3.3. Bone Mass, LBM, and Fat Mass Asymmetry Index

Asymmetry indices for bone mass, LBM, and fat mass in male and female soccer players are presented in Table 4. Refer to Figure 1 for asymmetry indices % for upper limb, trunk, and total LBM. There were no significant differences between male and female athletes’ AI regarding bone mass (i.e., upper limb, lower limb, trunk, and total) and LBM (i.e., upper limb, lower limb, trunk, and total). However, there was a significant difference in the upper limb fat mass AI in male soccer players (11.06%) compared to female soccer players (3.56%). Additionally, there was a significant difference in the trunk fat mass AI in male soccer players (9.86%) compared to female soccer players (5.36%). There was a significant difference in the trunk total fat mass AI in male soccer players (5.90%) compared to female soccer players (1.55%). 

### 3.4. Asymmetry Index Odds Ratio in Male and Female Soccer Players

The logistic regression analysis (odds ratio) for the AI, LLI, and LBP is presented in Table 5. No significant findings were found; however, the greater lower limb LBM AI (OR = 1.42, 95% CI = 0.99–2.03, *p* = 0.053) and trunk LBM AI (OR = 1.33, 95% CI = 0.98–1.81, *p* = 0.069) approached significance as possible predictors of LLI in the last 12 months and LLI in the last 4 months, respectively. 

## 4. Discussion

### 4.1. Lower Limb LBM Asymmetry 

The males in our sample had exceptionally low lower limb LBM AI (3.82%) compared to the normal range (10–25%) (Refer to Table 1). The females also had low lower LBM AI (3.36%) compared to the normal range (5–13%) (Refer to Table 1). These findings are consistent with a previous study of a similar design [4], suggesting collegiate soccer players do not have significant side-to-side asymmetries in lower limb LBM. Many factors could explain why we observed symmetry in lower limb LBM. The first reason can be better understood by analyzing the demands of soccer. Running involves constant bilateral forces, and most likely outweighs the repetitive unilateral motion of kicking [4]. In game settings, soccer players spend most of their time running without the ball. Studies looking at professional soccer players found during a 90 min match, players only have possession of the ball for 81 s on average [32] and touch the ball approximately 39.6 times [33]. In contrast, soccer players typically run about 10 kilometers during a 90 min match, spending 80–90% of their time near their anaerobic threshold [34]. A systematic review looking at elite soccer players found they run at very high speeds for approximately 1000 m and sprint for 300 m during a 90 min match [35]. Therefore, soccer players spend a lot more time exerting effort running rather than handling or kicking the ball. The symmetrical demand of running at different speeds and intensities over extended periods likely has a more significant impact on maintaining symmetry than unilateral kicking. 

Another potential factor is the long-term training adaptation that occurs in elite soccer players [4]. Elite soccer players minimize the unilateral impact of their sport through routine training versatility and incorporating a variety of exercise modalities. In our sample, the average female soccer player played at a competitive level for 8.8 ± 2.6 years with 1.6 ± 1.2 years played at a university level, while the average male soccer player played at a competitive level for 8.3 ± 3.5 years with 1.3 ± 1.4 years played at a university level. Athletes in our sample had a lot of competitive experience, which may yield better long-term training adaptations. The first training adaptation often arises from proper strength and conditioning. Strength and conditioning is an important part of the periodization training plan for competitive soccer players, especially at a university level. A properly executed strength and conditioning plan can reduce asymmetries and improve athletic performance [36]. The second component of long-term training adaptation can occur due to the competitive advantage of being two-footed. Being two-footed refers to a player’s ability to kick, pass, and tackle with their non-dominant leg. Success at high levels requires adequate control of the non-dominant leg [37]. Two-footedness is a critical skill to master to control the ball while running, allowing players to quickly react and adapt during play, thus giving them a strategic advantage. The type of training players receive to promote two-footedness combined with the years of strength and conditioning experience they receive likely contributes to reducing asymmetries. 

### 4.2. Upper Limb and Trunk LBM Asymmetry

Males and females both exceeded their respective thresholds for the high upper limb LBM AI (males 7.3% > 7%; females 4.39% > 3%) [29]. Additionally, males had a trunk LBM AI of 4.49%, while females had a trunk LBM AI of 4.24%. There are no normative data for trunk LBM AI; however, by comparing the indices to the average body composition measurements from Table 3, we can infer these differences are somewhat elevated. Males had a trunk LBM difference that neared significance at *p* = 0.08, as did females at *p* = 0.07. These findings are coherent with a previous study that found asymmetrical hypertrophy of the rectus abdominis [26]. Delang et al. indirectly suggested upper limb LBM asymmetry could be explained by a compensation mechanism for dominant limb proficiency [4]. The compensation mechanism relates to the biomechanics of kicking and the transmission of load through the body. The biomechanics of kicking are not limited to the lower extremity but rather involve the entire body. The transmission of force begins from the spine before moving down into the pelvis and hips [38]. The backswing phase of kicking requires rotation and extension of the spine. The follow-through phase requires forceful flexion and opposite rotation of the spine [39]. The transition from the backswing phase to the follow-through phase requires tremendous stability of the spine to counteract the forceful rotation of the body. The force distribution could be unevenly dispersed throughout the body and primarily absorbed by the spine and its associated musculature. Consequently, the upper limb may experience asymmetrical force production, potentially leading to the development of upper limb LBM asymmetry over time. These factors could contribute to the high percentage of LBP seen in soccer players [7].

### 4.3. Asymmetry and Injury Risk

The logistic regression analysis did not yield any significant correlations. However, our findings suggest that a greater lower limb LBM AI and trunk LBM AI may be possible predictors of LLI, as both variables approached significance and could be potential risk factors of injury. It is possible that our study was underpowered to detect these associations; additional studies are warranted to confirm and expand our findings. 

### 4.4. Sex Differences

According to our normative data from Table 1, females generally tend to have lower LBM AI. However, our findings suggest there are no significant differences in LBM AI between males and females. This could be because we had a homogenous sample of highly trained soccer players that are not representative of the untrained population, which makes up most of the normative data. We observed a significantly greater arm, trunk, and total fat mass asymmetry in males compared to females. Males and females typically exhibit distinct patterns in body fat distribution. Females usually store more fat in the hip and thigh regions while males store more fat in the trunk and upper body regions [40]. This variation in body fat distribution could partly account for the differences observed in fat mass asymmetry, although the mechanism remains unclear. 

### 4.5. Limitations

There are limitations to this study, including the relatively small sample size and the focus on varsity soccer players, which may limit the generalizability of the findings to other populations, such as amateur players, various age groups, or individuals with varying skill levels. Furthermore, the small sample size may have limited our ability to detect other significant outcomes in terms of injury risk. Additionally, we did not account for limb dominance in the analysis seen in Table 2, but rather categorized based on left and right. However, 91.6% of females and 71.4% of males were right limb dominant, therefore the impact on the results was minimal. 

### 4.6. Future Studies

Future studies in the field of body composition asymmetry should focus on establishing injury risk cut-off values for LBM AI specific to an athletic population. Additionally, incorporating longitudinal studies may provide a more dynamic understanding of how asymmetries evolve over time in response to training regimens and competitive demands. Furthermore, examining the potential influence of player positions and playing styles on limb-dominance-related asymmetries could offer valuable insights into position-specific demands that may contribute to the observed variations. As the field progresses, collaborative efforts between researchers and practitioners may facilitate the development of targeted training interventions to address and optimize limb symmetry in varsity soccer players, ultimately enhancing performance and minimizing the risk of injuries associated with imbalances. 

## 5. Conclusions

The male and female soccer players in our sample displayed low lower limb LBM asymmetry. However, both males and females displayed high upper limb LBM asymmetry. We found no significant correlation between any AI and LLI or LBP, likely due to sample size limitations. Our findings suggest lower limb LBM asymmetry and trunk LBM asymmetry may be potential risk factors for LLI. Our findings are consistent with previous morphological studies conducted on soccer players and contribute to a deeper understanding of the effects of soccer on body composition symmetry. Nonetheless, further research is needed to confirm our findings. Exploring the suggested mechanisms for mitigating the impacts of unilateral kicking, notably through the adoption of diverse training routines and incorporating a range of exercise modalities, warrants deeper investigation and integration into training environments. While evaluating an athlete’s body composition for asymmetries holds promise for informing strength and conditioning programs and injury prevention strategies, it is crucial to recognize that lateral dominance in soccer may be influenced by multifaceted factors within the sport, warranting additional in-depth exploration.

## Figures and Tables

**Figure 1 ijerph-21-00559-f001:**
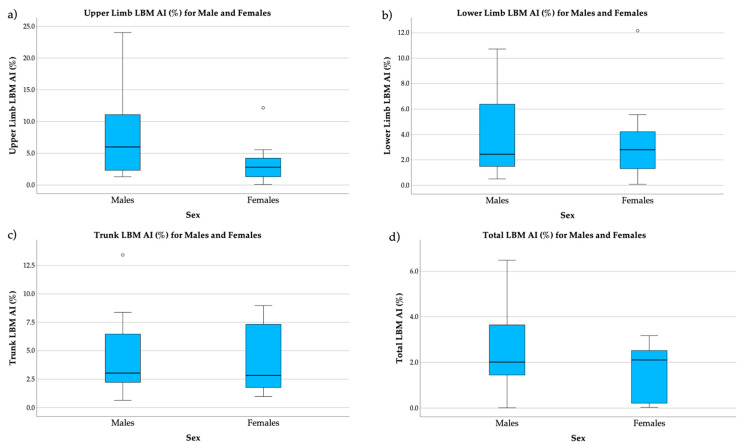
LBM AI in males and females: (**a**) upper limb LBM AI (%) in males and females; (**b**) lower limb LBM AI (%) in males and females; (**c**) trunk LBM AI (%) in males and females; (**d**) total LBM AI (%) in males and females. LBM: lean body mass; AI: asymmetry index; ° represents potential outliers.

**Table 1 ijerph-21-00559-t001:** Upper and lower limb LBM asymmetry index thresholds.

	Males	Females
Upper Limb LBM AI		
Low	3–7%	1–3%
High	>7%	>3%
Lower Limb LBM AI		
Low	10–25%	5–13%
High	>25%	>13%

LBM: lean body mass; AI: asymmetry index.

**Table 2 ijerph-21-00559-t002:** Participants’ demographic characteristics.

	Female (*n* = 12)	Male (*n* = 15)
Age (years)	20.5 ± 1.6	20.3 ± 1.9
Height (cm)	163.4 ± 8.5	179.5 ± 7.4
Weight (kg)	64.6 ± 8.2	72.1 ± 7.7
Total LBM (kg)	43.61 ± 4.1	59.1 ± 6.5
Total bone mass (kg)	2.6 ± 0.3	3.4 ± 0.4
Total body fat %	18.6 ± 5.7	10.0 ± 2.3
BMI (kg/m^2^)	24.3 ± 3.5	22.4 ± 1.8
Dominant leg (*n*)		
Right	11	11
Left	1	3
Either	0	1
Soccer competitive level (year)	8.8 ± 2.6	8.3 ± 3.5
Soccer university level (year)	1.6 ± 1.2	1.3 ± 1.4
LBP in past 3 months (n)	4	4
LBP intensity (NPRS)	3.6 ± 1.9	5.0 ± 1.6
LLI in past 12 months (n)	9	4

LBM: lean body mass; BMI: body mass index; LBP: low back pain; LLI: lower limb injury.

**Table 3 ijerph-21-00559-t003:** Total body composition results (bone mass, LBM, and fat mass).

	Males (*n* = 15)			Females (*n* = 12)		
	Right (Mean ± SD)	Left (Mean ± SD)	*p*-Value [95%CI]	Right (Mean ± SD)	Left (Mean ± SD)	*p*-Value [95%CI]
Bone Mass (g)						
Upper Limb	228.10 ± 36.83	214.87 ± 30.18	0.02 [2.56, 23.90] *	158.80 ± 25.64	153.25 ± 20.94	0.02 [0.98, 10.13] *
Lower Limb	685.22 ± 107.82	687.56 ± 95.90	0.62 [−12.13, 7.46]	481.98 ± 69.78	395.74 ± 69.78	0.19 [−12.36, 2.69]
Trunk	503.48 ± 82.35	506.56 ± 83.10	0.71 [−20.46, 14.29]	395.74 ± 60.76	400.47 ± 65.32	0.22 [−12.71, 3.25]
Total	1721.01 ± 245.72	1658.23 ± 202.55	0.04 [3.45, 122.08] *	1273.18 ± 181.17	1294.23 ± 161.22	0.26 [−60.37, 18.28]
LBM (g)						
Upper Limb	3611.97 ± 673.13	3421.02 ± 532.90	0.04 [10.06, 371.84] *	2305.08 ± 259.88	2252.18 ± 253.37	0.11 [−14.23, 120.04]
Lower Limb	10437 ±1342.73	10,404.61 ± 1104.41	0.82 [−280.76, 347.35]	7575.65 ± 931.29	7477.26 ± 894.35	0.35 [−122.30, 319.09]
Trunk	13,740.67 ± 1561.86	13,990.86 ± 1625.99	0.08 ^	10,385.01 ± 1040.08	10,665.40 ± 996.66	0.07 [−585.96, 25.19]
Total	29,678.98 ± 3316.42	29,383.10 ± 3234.16	0.24 [−215.09, 806.86]	21,675.83 ± 2138.11	21,936.14 ± 2013.13	0.02 [−478.31, −42.31] *
Fat Mass (g)						
Upper Limb	555.92 ± 141.19	521.35 ± 120.45	0.09 [−5.69, 74.84]	1065.76 ± 295.76	1028.50 ± 277.12	0.01 ^*
Lower Limb	1818.30 ± 564.08	1854.55 ± 526.66	0.13 [−84.79, 12.30]	3696.04 ± 1002.07	3640.08 ± 924.03	0.94 ^
Trunk	2159.85 ± 597.5	2232.40 ± 691.23	0.36 ^	4151.76 ± 1658.56	4213.42 ± 1695.51	0.41 [−281.38, 95.25]
Total	5028.56 ± 1108.52	5015.95 ± 1250.35	0.90 [−188.14, 213.35]	9292.22 ± 2869.13	9294.82 ± 2832.76	0.97 [−131.35, 126.15]

*: significant *p*-value *p* < 0.05. ^: Wilcoxon signed rank test was performed, no 95% CI available. LBM: lean body mass.

**Table 4 ijerph-21-00559-t004:** Asymmetry index for bone mass, LBM, and fat mass in soccer players.

	Male	Female	|Difference|	*p*-Value [95% CI]
	Asymmetry Index (%)			
Bone Mass				
Upper Limb	7.60 ± 6.86	4.35 ± 2.88	3.25	0.14 [−7.65, 1.15]
Lower Limb	1.85 ± 1.87	2.29 ± 1.59	0.44	0.50 [−0.98, 1.86]
Trunk	4.70 ± 4.01	2.80 ± 1.91	1.90	0.15 [−4.52, 0.72]
Total	5.74 ± 4.01	4.24 ± 3.22	1.50	0.31 [−4.49, 1.48]
LBM				
Upper Limb	7.30 ± 6.51	4.39 ± 2.51	2.91	0.16 [−7.03, 1.22]
Lower Limb	3.82 ± 3.39	3.36 ± 3.24	0.46	0.73 [−3.16, 2.24]
Trunk	4.49 ± 3.42	4.24 ± 3.10	0.25	0.85 [−2.91, 2.41]
Total	2.50 ± 1.84	1.58 ± 1.16	0.92	0.15 [−2.19, 0.35]
Fat Mass				
Upper Limb	11.06 ± 9.09	3.56 ± 3.34	7.50 *	0.01 [−13.22, −1.80]
Lower Limb	4.98 ± 5.19	3.86 ± 3.82	1.12	0.54 [−4.88, 2.62]
Trunk	9.86 ± 4.96	5.36 ± 4.39	4.50 *	0.02 [−8.32, −0.68]
Total	5.90 ± 4.22	1.55 ± 1.15	4.35 *	0.002 [1.74, 6.94]

*: significant *p*-value *p* < 0.05. LBM: lean body mass.

**Table 5 ijerph-21-00559-t005:** Logistic regression analysis for asymmetry index and LLI and LBP risk.

Asymmetry Variable	Odds Ratio	95% CI	*p*-Value
Left–Right lower limb AI × LLI (12 months)	1.46	0.85–2.51	0.170
Lower limb AI × LLI (12 months)	1.42	0.99–2.03	0.053
Trunk bone mass AI × LLI (4 weeks)	1.34	0.95–1.90	0.096
Trunk LBM AI × LLI (12 months)	1.33	0.98–1.81	0.069
Trunk LBM AI × LLI (3 months)	1.27	0.93–1.74	0.135
Total fat mass AI × LBP (3 months)	1.24	0.91–1.67	0.171
Total bone mass AI × LLI (12 months)	1.23	0.93–1.61	0.144
Total LBM AI × LLI (12 months)	1.22	0.70–2.15	0.483

AI: asymmetry index; LLI: lower limb injury; LBM: lean body mass; LBP: low back pain.

## Data Availability

All extracted data is available upon request.

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
