# Peer review of "DEXA Body Composition Asymmetry Analysis and Association to Injury Risk and Low Back Pain in University Soccer Players"

_ijerph, 2024, doi:10.3390/ijerph21050559_

Round 1
Reviewer 1 Report (Previous Reviewer 2)
Comments and Suggestions for Authors
First of all, thank you for this study that will contribute to the literature. The title of the study can be changed from "Soccer Athletes" to "Soccer Players". I found grammatical errors in some paragraphs and I have added them as explanations. These can also be corrected. Apart from these, the selected statistical analyses were used appropriately. Introduction, Method, Discussion and Conclusion are sufficiently explained. In some paragraphs, it is stated as "university football player". After all, they are football players. It can also be stated as "football player". In the method, it is sufficient to say university football players. I think it is not necessary in other paragraphs.

Comments on the Quality of English LanguageIn fact, there are grammatical errors in the gas, I think support can be obtained.
Author Response
Please see the attachment.

Reviewer 2 Report (New Reviewer)
Comments and Suggestions for Authors
Dear Authors,
Special thanks for your manuscript submission and to the editors for providing the opportunity to review this work.
The study, titled “DEXA Body Composition Asymmetry Analysis and Association to Injury Risk and Low Back Pain in University Soccer Athletes”,' aims to 1) examine body composition asymmetries in university-level soccer players, and 2) determine if body composition asymmetry is associated with LLI and LBP.
The manuscript is engaging, and here are some insights and suggestions for the methodology and subject area:
- The procedures and experimental protocol are clear and well-written.
- Relocate participant details and descriptions to a dedicated sub-section (Participants) in the material and methods section (lines 186-195).
- The results are clear and well organized.
- The discussion and conclusions are good and address the aim of the study, but need improvement for more clarity.
- The references used in this study are relevant and suitable.
- Finally, thanks to the authors' effort, this scientific work is very interesting and beneficial to the reader.
Overall, you had an interesting manuscript. The main question addressed by the research is a relevant topic for basic research in the field therefore it holds potential interest for IJERPH's readership.
Author Response
Please see the attachment.

This manuscript is a resubmission of an earlier submission. The following is a list of the peer review reports and author responses from that submission.
Round 1
Reviewer 1 Report
Comments and Suggestions for Authors
The paper discusses the risks of pain and injury due to asymmetries in soccer players. While the authors present interesting findings related to DEXA body composition asymmetry analyses, several aspects should be deepened to improve the work:
1. Introduction: the authors do not mention that early stage injuries may not appear at the DEXA-level (that corresponds to the meso-scale). Indeed, it is well recognized that early damage start appearing at lower scales (up to the micro-scale). It is worth discussing this aspect in the paper (see other literature works: https://doi.org/10.1038/nmat1866 , https://doi.org/10.1016/j.mtla.2021.101229 )
2. Materials and methods: the authors did not show a sample size analysis. Indeed, it should be included to demonstrate the reliability of the obtained observations.
3. Results: to facilitate the readability of the results, boxplots may be included in addition to Table 2, 3 and 4.
Reviewer 2 Report
Comments and Suggestions for Authors
First of all, I should state that the number of samples is very small, and reaching a conclusion and generalising with this number will not give the right result. I also detected some grammatical errors in the study and your hypothesis is not very clear. Having 8 years of football background is also an advantage. A group with more football player background could have been selected.

Comments on the Quality of English LanguageThe English language of the article should be improved.